# Combination of mTOR and MAPK Inhibitors—A Potential Way to Treat Renal Cell Carcinoma

**DOI:** 10.3390/medsci4040016

**Published:** 2016-10-15

**Authors:** Ashutosh Chauhan, Deepak Kumar Semwal, Satyendra Prasad Mishra, Sandeep Goyal, Rajendra Marathe, Ruchi Badoni Semwal

**Affiliations:** 1Department of Urology, Postgraduate Institute of Medical Education & Research, Chandigarh 160012, India; 2Department of Biotechnology, Faculty of Biomedical Sciences, Uttarakhand Ayurved University, Harrawala, Dehradun 248001, Uttarakhand, India; 3Department of Phytochemistry, Faculty of Biomedical Sciences, Uttarakhand Ayurved University, Harrawala, Dehradun 248001, Uttarakhand, India; 4Vice Chancellor, Uttarakhand Ayurved University, Harrawala, Dehradun 248001, Uttarakhand, India; vc@uau.ac.in; 5Department of Hepatology, Postgraduate Institute of Medical Education & Research, Chandigarh 160012, India; sandeepgoyalpgi@gmail.com; 6Department of Pediatrics, Postgraduate Institute of Medical Education & Research, Chandigarh 160012, India; rajendra.bt09@gmail.com; 7Department of Pharmaceutical Sciences, Tshwane University of Technology, Pretoria 0001, South Africa; semwalrb@tut.ac.za

**Keywords:** renal cell carcinoma, mammalian target of rapamycin, mitogen activated protein kinase, target therapy

## Abstract

Renal cell carcinoma (RCC) is the most common neoplasm that occurs in the kidney and is marked by a unique biology, with a long history of poor response to conventional cancer treatments. In the past few years, there have been significant advancements to understand the biology of RCC. This has led to the introduction of novel targeted therapies in the management of patients with metastatic disease. Patients treated with targeted therapies for RCC had shown positive impact on overall survival, however, no cure is possible and patients need to undergo treatment for long periods of time, which raises challenges to manage the associated adverse events. Moreover, many patients may not respond to it and even response may not last long enough in the responders. Many inhibitors of the Mammalian target of Rapamycin (mTOR) signaling pathway are currently being used in treatment of advanced RCC. Studies showed that inhibitions of mTOR pathways induce Mitogen-Activated Protein Kinase (MAPK) escape cell death and cells become resistant to mTOR inhibitors. Because of this, there is a need to inhibit both pathways with their inhibitors comparatively for a better outcome and treatment of patients with RCC.

## 1. Introduction

Renal cell carcinoma (RCC) accounts for almost 3% of all type of cancers, with approximately 116,000 deaths occurring per year worldwide [1]. RCC is often diagnosed at its metastatic stage and has a great resistance to chemotherapy as well as to radiotherapy, which makes up for its poor prognosis [2]. Favorable response to treatment is observed only in a very small subset of patients who are given immunotherapy [3]. Once RCC has metastasized, survival rate becomes less than 10% at 5 years [4]. A large number of RCC related deaths occur only due to poor detection of disease. Various biomarkers that have been evaluated in RCC are used for diagnostic, predictive and prognostic purposes. Among these markers, the main emphasis has been given to prostate specific membrane antigen (PSMA) and carbonic anhydrase (MN/CA-9). However, a study has indicated that MN/CA9 and PSMA have high specificity but low sensitivity in blood samples of patients with RCC [5]. Thus, RCC treatment needs to be improved by inhibiting two major pathways in cell survival, i.e., Mitogen-activated protein kinases (MAPKs) and Mammalian target of Rapamycin (mTOR).

MAPKs are related to a family of serine/threonine protein kinases, and are extensively conserved amongst eukaryotes. They are involved in a number of cellular processes like proliferation, death and differentiation. Mutations in proteins involved in these pathways contribute to nearly 20% of all human cancers [6]. Various earlier researches already suggested a differential role of MAPKs in cancers, where they have a dual role, assisting in carcinogenesis in RCC and acting as tumor suppressors in gastric adenocarcinoma [7,8]. However, this differential behavior may depend on the cell types, different environmental stimuli and the genetic constitution of individual cells and organisms.

mTOR pathway has also been shown to be upregulated in RCC and is the second major pathway targeted by current therapeutic options. Moreover, alteration at the level of mTOR activity promotes the invasiveness and metastatic potential in a variety of malignancies such as glioblastoma, small cell lung cancer, osteosarcoma, pancreatic cancer, leukemia, breast, colon cancer and RCC [9]. Therefore, the combined effects of mTOR inhibition with p38 MAPK inhibition can help in the treatment of RCC by inducing apoptosis and decreasing cell proliferation.

The purpose of the present review is to explore the current status, particularly pathogenesis, diagnosis and treatment of RCC. The literature was imported from PubMed and Scopus data base by the search terms “mTOR inhibitor and RCC”, “MAPK inhibitor and RCC”, “combination of mTOR and MAPKs inhibitors”, “renal cell carcinoma and clinical trial”, etc. Around 500 relevant papers were reviewed in which only the most relevant reports were considered. This review looks at what is known about the prevalence of RCC to challenges in its treatment, and solutions with the help of combination target therapy.

## 2. Renal Cell Carcinoma

The incidence of RCC varies geographically, and is found to be the highest in Europe, North America, as well as Australia; whereas in Asia, its incidence is relatively low. Worldwide incidence of RCC is increasing by about 2% annually. In 2008, about 88,400 new cases of kidney cancer were identified in Europe [10] making it the 10th most common cancer. Except for few European countries such as Sweden, Poland, Finland, and the Netherlands, an increase in RCC has been observed worldwide during the past few years [11].

Despite the increased incidence of RCC, the survival rate of patients with RCC has been also shown to improve in the last five years due to its early diagnosis, and the economical as well as approachable treatment modalities. Even in conditions such as distant metastasis, 5-year survival rates have increased more than 10% [12]. In this new medical era with improved diagnosis and imaging techniques, the survival rate is increasing among patients. Tumors are being detected at an earlier stage as well as with a smaller size [13].

Till date, there is no exact calculable data available regarding the incidence and prevalence of RCC in the Indian population. On average, 160–190 new cases of urological malignancies including 50–60 cases of RCC are managed annually at Postgraduate Institute of Medical Education and Research (PGIMER), Chandigarh (India). However, very few studies have been conducted in the remaining parts of India with respect to epidemiological data [14], radiology and imaging [15], tumor markers in RCC [16,17] and prevalence of RCC and its histopathological subtype [18]. A clinicopathological study of 388 tumors of urogenital tract conducted by Sharma et al. [19] revealed that 376 tumors (97%) were malignant and the remaining 12 (3%) were benign. Renal tumors constituted 10.6% of all the urogenital tumors. Khaitan et al. [14] reported that, out of 400 cases of RCC, 144 cases had tumor in stage I, 68 had stage II, 99 had stage III and 119 had stage IV tumors. Singh et al. [16] evaluated the role of serum transforming growth factor-β_1_ (TGF-β_1_) and MN/CA9 as tumor markers in RCC. The authors reported that TGF-β_1_ levels were significantly higher in patients with RCC compared to controls. Additionally, there was a significant decrease in serum TGF β_1_ following nephrectomy. Sharma et al. [19] also found that cancer cells with MN/CA 9 are present in circulation in 68% of cases with RCC. Prasad et al. [17] studied alteration in expression and activity of alkaline phosphatase in brush border membrane in RCC, and suggested that alteration in expression could help to better understand the molecular basis of the pathophysiology of RCC as well as the development of new tumor markers. Pradhan et al. [18] reported that clear cell carcinoma was the most prevalent (74.8%), followed by papillary (12.2%), chromophobe (7.9%), and oncocytoma (1.8%); 1 case of collecting duct and 8 cases of sarcomatoid of RCC out of 278 of cases in the last 10 years were reported. RCC has been considered as a unique and challenging cancer because of paraneoplastic syndrome including hypercalcemia, erythrocytosis and nonmetastatic hepatic dysfunctioning (i.e., Staffer syndrome) and poor outcome in advanced cases. This cancer is characteristically resistant to conventional oncological therapies such as chemotherapy, radiotherapy and immune therapy [20], hence making it difficult to successfully treat through therapeutic approaches. Surgery has remained the major therapeutic option. Despite the tremendous progress made in medical science, the pathogenesis of RCC is still only partially known. However, the abnormalities of the von Hippel-Lindau (VHL), a tumor suppressor, have been implicated in the majority of RCC cases.

## 3. Etiology: Potential Risk Factors for Renal Cell Carcinoma

There are various risk factors associated with RCC, among them smoking, alcohol consumption, unhealthy diet and heredity factors are important, and their possible role in RCC is confirmed by some previous researches. The possible risk factors and their roles in RCC are given in Table 1.

## 4. Classification, Staging and Grading

RCC is assessed according to Heidelberg, the Union for International Cancer Control (UICC) and the American Joint Committee on Cancer (AJCC) classification [44,45]. RCC has traditionally been classified according to cell type and growth pattern. This classification gives little insight into the clinical behavior of the carcinoma. Histological subtypes of RCC according to the Heidelberg classification are divided into Clear cell carcinoma (cRCC), Papillary (pRCC), Chromophobe (chRCC) and Collecting duct carcinoma (cdRCC).

### 4.1. Clear Cell Carcinoma

cRCC is found in 85% of cases [46]. Dalgliesh et al. [47] sought to identify additional genetic alterations in RCC. Overall, cRCC has fewer mutations than other common solid tumors. cRCC is characterized by loss of genetic material on chromosomes 3, 8, 9 and 10. Approximately 50% of cRCCs have somatic mutations in the VHL gene and, moreover, 10%–20% of these tumors show inactivation of VHL gene [48]. Loss of heterozygosity on chromosome 8p or 9p has prognostic significance for cRCC progression. However, these molecular biomarkers are yet to gain general use in RCC diagnosis and prognosis. Studies have identified some genes that might be useful for prognostic purposes or for the classification of RCC [49]. The largest percentage of genetic aberrations that are associated with cRCC includes 3p loss or changes [50].

### 4.2. Papillary Renal Cell Carcinoma

pRCC, also known as chromophilic RCC, is another common subtype accounting for 10%–15% of all RCC [51]. In such cases, a papillary growth pattern predominates, although tubule-papillary and solid architecture may be seen. pRCC is a further sub group, grouped either as type 1 with small cell size and pale cytoplasm as the characteristics, or type 2 whereby cells are larger with eosinophilic cytoplasm and pseudostratified nuclei [52].

### 4.3. Chromophobe Rencal Cell Carcinoma

Approximately 5% of all RCC cases belong to chRCC in which cells comprise many micro vesicles and have pale or eosinophilic granular cytoplasm. The cytoplasm is condensed near the cell membrane and produces a halo around the nucleus [44]. These tumors are slow in growth and occasionally metastasize leading to good prognosis. chRCC are characterized by monosomy of multiple chromosomes, namely 1, 2, 6, 13, 17, and 21 [50].

### 4.4. Collecting Duct Carcinoma

Collecting duct carcinoma and medullary carcinoma comprise around 1% of all RCC and are usually quite aggressive. Medullary carcinoma is sometimes considered as a variant of collecting duct carcinoma and is associated with sickle cell trait or disease [53]. These are characterized by an irregular channel lined by highly atypical epithelium. There are no genetic abnormalities associated with such cases [44].

### 4.5. Unclassified Renal Cell Carcinoma

The cases which do not fit into any of the above mentioned known categories are said to be unclassified. This subtype accounts for 3%–5% of total tumors in surgical series. Since this category contains tumors with a diversity of appearances and genetic lesions, they are not subject to a limiting definition. The main features are Sarcomatoid morphology without recognizable epithelial elements, mucin production, mixtures of epithelial and stromal elements, and unrecognizable cell types [44].

### 4.6. Tumor-Node-Metastasis Staging

Staging is the method to determine the spreading of cancer in the body and the place where it is residing. Staging describes the severity of an individual’s cancer based on the spreading of the original (primary) tumor in the body. The tumor-node-metastasis (TNM) staging system is the most commonly used staging system. This system was developed and is maintained by the AJCC and the UICC (Table 2, Table 3, Table 4 and Table 5).

### 4.7. Nuclear Grade

Skinner et al. [54] were the first to propose a grading system based solely on nuclear morphology in 1971 which was later simplified by Fuhrman et al. in 1982 [55]. Currently, the Fuhrman nuclear grading is one of the most widely used systems for grading RCC [56] (Table 6). This system is based on nuclear size, shape and prominence of nucleoli [57].

## 5. Diagnosis and Treatment

RCC in its early stage is relatively asymptomatic; however, few symptoms such as flank pain, hematuria and renal mass are manifested at a very late stage. The diagnosis of RCC is further confirmed with imaging studies such as computerized tomography (CT) scan and ultrasound and, in many cases, RCC is found incidentally during routine imaging [53]. Detection in early stages and the recent development of treatment options have positively influenced the prognosis for patients with this disease. As spreading and treatment responses are still not predictable, hence, molecular markers could be helpful in refining individual risk stratification and treatment decisions.

## 6. Molecular Markers

Each of these diagnostic techniques provide unique information regarding tumor size and extent of RCC, but these methods diagnose disease at advanced stages, when tumor becomes metastatic. There has been an increased understanding of the molecular characteristics and hereditary forms of RCC, which has led to further elucidation of pathways involved in RCC. Many other promising new molecular techniques including microarray analysis, single nucleotide polymorphism, gene expression profiling, proteomics and improved bioinformatics have increased our ability to identify molecular and genetic abnormalities and help in our search for potential biomarkers. To date, no standard approaches to biomarker sampling or analysis have been adopted for RCC diagnosis as many of the putative tumor markers themselves are still under active investigation. Some of them are listed below.

### 6.1. Diagnostic Biomarkers

Diagnosis of RCC in early stages remains a challenge for oncologists and presents a barrier to reduce the mortality of RCC. Although in recent years increased incidental detection of small renal masses has been seen, the mortality rate remains the same, suggesting that renal cancers with lethal potential are not identified sufficiently early to prevent metastatic spread. The diagnosis of RCC using biomarkers presents a major opportunity to reduce RCC related death. Some of those diagnostic biomarkers are as follows.

#### 6.1.1. MN/CA9 and Circulating Cell Detection Biomarkers

The tumor-associated antigen MN/CA9 was first described as a cell surface protein on the HeLa cell line. Further mRNA expression of MN/CA9 was detected in cRCC cases whereas normal renal parenchyma from tumor-bearing kidneys had lack of mRNA expression [5]. There are two different molecular markers, MN/CA9 [58] and PSMA [5] that have been reported in RCC, which were also detected in blood samples of patients with RCC. Their expressions were also found to be correlated with vascular invasion.

#### 6.1.2. Urinary Biomarkers

A recent study by Morrissey et al. [59] has shown that aquaporin-1 and adipophilin concentrations in patients with a pathologic diagnosis of clear cell or papillary cancer are significantly increased in patients with renal cancer of nonproximal tubular origin. Collectively, these proteins account for about 90% of all kidney cancers. The concentrations of the above proteins were correlated with size of tumor and found that the concentrations decreased sharply after nephrectomy.

#### 6.1.3. Serum Nucleic Acids

Micro ribonucleic acids (miRNAs) are short non-coding nucleotides that are associated with some cases of malignancies [60]. Although their exact role in carcinogenesis is still not clear, many of them have been linked with RCC. miR-210, a miRNA regulated by the hypoxia inducible factor 1 alpha (HIF-1α) pathway, was found to have significantly higher levels in cRCC tissue as compared with normal renal parenchyma and higher levels in serum of patients with cRCC as compared to healthy controls [61]. Other groups also have shown that miR-378, miR-451, and miR-1233 levels are distinctly elevated in patients with RCC [62,63]. A recent study has shown that the absolute level of serum cell-free deoxyribonucleic acid and the level of CpG island methylation of the Ras association domain family 1 isoform A (RASS F1A) and VHL loci were higher in patients with RCC compared with controls [64]. Although these studies are promising in small cohorts of patients, they need to be replicated and validated in large populations.

#### 6.1.4. Composite Biomarkers

There are some limitations with individual blood-based biomarkers, so investigators have made an attempt to use a combination of multiple biomarkers thought to be associated with RCC in the hope of obtaining better diagnosis than individual biomarkers themselves. Recently, the combination of nicotinamide N-methyltransferase (NNMT), L-plastin (LCP1), and nonmetastatic cells1 (NM23A) protein were found to be elevated in serum samples of RCC patients [65]. A composite index of concentrations of all three yielded an area under curve (AUC) of 0.932. External validation done on a test data set yielded an AUC of 0.919. Although preliminary results are encouraging, there is a need for further validation in large groups of patients.

### 6.2. Prognostic Biomarkers

Prognostication in RCC started long ago. One of the earliest studies used multivariate modeling to predict cancer specific mortality of metastatic RCC using clinical variables like Karnofsky performance status, lactate dehydrogenase, hemoglobin concentration in blood, calcium, and whether a nephrectomy was performed [66]. Further refinements have been made with advanced urinary cancers of approximately 80%–85% [67,68]. These models are far from perfect and it is believed that future refinements need to be made with the identification of more specific molecular markers. Some of the most common and recent prognostic markers for RCC are given below.

#### 6.2.1. Proliferative Index

Mitotic index is the number of mitosis per field counted under the microscope. There is a significant association of tumor mitotic rate with survival of RCC [69]. Argyrophyllic nucleolar protein count also has been correlated with tumor grading and malignancy. However, its utility as a predictor biomarker is yet to be established in RCC.

#### 6.2.2. Proliferative Biomarker

Ki-67, a nuclear antigen, occurs throughout the active phases of the cell cycle (G1, S, G2, and M) except in resting G0 phase or early G1 phase. The expression of Ki-67 protein has been correlated with proliferative index, biological aggressiveness, tumor cell growth, and prognosis of several cancers, including RCC [70]. Similar to Ki-67, proliferating cell nuclear antigen expressed during the replicative phase is used as a marker of dividing cells. Such biomarkers permit stratification of patients with localized tumors who are likely to develop recurrence after nephrectomy and predict their survival. Gayed et al. [71] have reported that cumulative number of aberrantly expressed biomarkers related to cell cycle and proliferation correlates with aggressive pathology of cRCC.

#### 6.2.3. Adhesion Molecules and Proteases

Although E-cadherin is a critical molecule for epithelium integrity, in most of the RCC cases its expression is found to be absent. Katagiri et al. [72] have reported that the prognostic value of E-cadherin is controversial in RCC. However, Shimazui et al. [73] found a significant correlation between survival and decreased expression of α-catenin.

Advanced stages of renal cancer have high mRNA expression of cathepsin L (protease), reported by Chauhan et al. [74]. Still, more detailed studies are required to validate the potential of cathepsin L as a prognostic biomarker. Neovascularization promotes rapid growth and metastatic dissemination of tumors by facilitating nutrient and metabolite exchange. In addition, intra-tumoral microvessel density is also reported to be highly indicative of survival in the initial stage of renal tumors [75].

#### 6.2.4. Apoptotic Regulatory Proteins

An internal membrane-associated protein, Bcl-2 oncoprotein, prolonged cell survival by inhibiting apoptosis. Bcl-2 expression has been found to be increased in RCC with the most pronounced immunoreactivity being observed in pRCC [76]. Survivin is another apoptosis inhibitor protein and is generally expressed during fetal development [77]. Its function is thought to be the inhibition of both the intrinsic and extrinsic caspase pathways. However, in normal adult tissues, it is quiescent but overexpressed in many cancers including RCC [78]. Studies suggested that increased survivin expression is associated with higher tumor stage and grade [79,80].

Some studies also supported the association of sarcomatoid transformation in renal cancer with the mutation of the p53 gene and p53 protein overexpression [81]. Furthermore, Shalitin et al. [82] described an enzyme linked immunosorbent assay (ELISA) for circulating p21 protein, an inhibitor of cyclin dependent kinases and a target of p53 which can act as a potential tumor marker for RCC. Recently, Nishikawa et al. [83] recommended combined evaluation of the expression levels of potential markers in the mTOR signaling pathway, which would contribute to the accurate prediction of disease recurrence following nephrectomies in RCC cases.

### 6.3. Predictive Biomarkers

Predictive biomarkers are used to test the expression of molecules in RCC patients prior to start the treatment as well as during the treatment to monitor the effect of line of treatment. The therapeutic landscape in locally advanced and metastatic RCC has changed dramatically in the past few years with the introduction of agents that target angiogenesis pathways including tyrosine kinase inhibitors (sunitinib, sorafenib, pazopanib, and axitinib), mTOR inhibitors (temsirolimus and everolimus) and direct vascular endothelial growth factor (VEGF) inhibitors (bevacizumab) [84]. These agents increased the number of treatment options, but, to date, no biomarkers exist for patient selection and to monitor the target response.

Studies have reported carbonic anhydrase IX (CAIX) expression as a predictive biomarker in response to interleukin-2 (IL-2) immunotherapy [85,86]. In a recent study, Tran et al. [87] have shown that pazopanib, IL-6, IL-8, VEGF, osteopontin, E-selectin, and hepatocyte growth factor (HGF) levels were associated with continuous tumor shrinkage with pazopanib treatment in phase 1 clinical trials. A similar study by Zurita et al. [88] reported osteopontin and VEGF at baseline predict the progression-free survival after treatment with sorafenib vs. sorafenib plus interferon-α (IFN-α) group in phase-2 trial. Although the above mentioned biomarkers were not proven to be potential biomarkers because most of them have either low sensitivity or low specificity, other newly discovered biomarkers need further validation in large groups of patients.

## 7. Treatments

RCC is one of the most chemotherapy and radiotherapy-resistant cancers of all human malignancies [89]. Furthermore, in a small proportion of cases, it does not even respond to immunotherapy with IL-2 or IFN-α, or even targeted therapy [89]. In early stage cases, ablation with cryotherapy and surgery may be attempted. Surgical resection is the most effective treatment for localized RCC tumors, but no satisfactory treatment is available for patients with advanced stage tumors. To date, several therapies for RCCs have not achieved a good response due to frequent occurrence of severe adverse reactions that does not allow for a satisfactory prognosis for patients. Although tumor staging is considered to be the most informative prognostic factor, up until today, no clear underlying molecular mechanism has been found which causes and leads to RCC [50]. If the cancer is confined only in the kidney (about 40% of cases), it can be cured with surgery (~90% of cases) but it often spreads to the lymph nodes or the main vein of the kidney, and it must be treated with adjunctive therapy including cytoreductive surgery.

### 7.1. Surgical Treatment

Surgery is the only curative treatment for localized RCC. Open radical nephrectomy is most frequently performed, although laproscopic and nephron/adrenal-sparing partial nephrectomy is gaining acceptance [53]. Nephectomy in metastatic cases has improved overall survival. Metastasectomy can be performed if there is single metastasis found after nephectomy [90,91].

### 7.2. Systemic Treatment

Medically, the treatment of advanced RCC is largely unsuccessful. Only 5%–10% of patients respond to chemotherapy [92,93]. The resistance to chemotherapy is probably due to the multidrug resistance gene (MDR-1) which is a possible neutralizer of cytotoxic agents and is commonly unregulated in renal tumors [94]. This may also slow down the unpredictable growth of RCC, which offers protection against chemotherapy [90]. Immunomodulatory therapy with cytokines is presently the main treatment regimen for RCC. Interferon-α and IL-2, have almost similar affectivity in the treatment of metastatic RCC [91]. However, recent data from Phase III suggested that cabozantinib and nivolumab may be new treatment options for patients with RCC, whereas phase II data suggest a combination of lenvatinib–everolimus may be promising options for the future [95].

Sorafenib and sunitinib, the inhibitors of VEGFR, PDGFR and FMS-like tyrosine kinase 3 (Flt3), are approved for the treatment of advanced RCC [91]. In addition, two mTOR inhibitors, temsirolimus and everolimus, were also approved by the Food and Drug Administration (FDA) in 2007 and 2009, respectively [96]. New target agent such as axitinib, pazopanib, cediranib, volociximab, tivozanib, and c-Met inhibitors such as GSK1363089 and ARQ197 are expanding treatment options. New sequential and combination targeted therapies are undergoing trials for advanced disease as adjuvant and neo-adjuvant approaches with nephrectomy [97].

## 8. Molecular Pathways in RCC

### 8.1. Mitogen-Activated Protein Kinase Pathway

Extracellular stimuli activate the MAPK pathways through mechanisms mediated by GTPases, including rat sarcoma (RAS), Ras-related C3 botulinum toxin substrate (RAC), cell division cycle 42 (CDC42) and Ras homologous (Rho). Once MAPKKKs (MAPK kinase kinases) are activated, they phosphorylate MAPKKs (MAPK kinases) on two serine residues. MAPKKs in turn phosphorylate the MAPKs such as the extracellular-signal-regulated kinase (ERK), JUN N-terminal kinase (JNK) and p38 on both threonine and tyrosine residues, which results in the catalytic activation of these MAPKs. Activated MAPKs then regulate transcription of various genes. ERK pathway is activated by growth factors, whereas stress and inflammatory cytokines preferentially activate the JNK and p38 pathways. Duration and strength of activation of the pathway is dependent on the sub cellular location of the variety of molecules involved in the pathway. These factors are involved in the activation of the pVHL, which is certainly the key element in the transformation process of the cRCC. In this regard, a study suggested that high-grade RCCs have higher MAPK activation than low-grade [7]. Therefore, activation of the MAPK cascade plays an important role in the carcinogenesis of RCCs. Later on, Samaras et al. [98] reported that p38 can serve as a potential biomarker of cRCC, which also correlates with Fuhrman grade. Another study has shown a significantly higher expression of MAPKK1 and ERK2 in cRCC patients relative to controls, suggesting that this pathway may have therapeutic value against cRCC [99]. A recent study has also shown that there is hyperactivation of JNK pathway which is independent of pVHL-deficiency in RCC [100]. There are many drugs used clinically to inhibit the MAPK pathway such as emurafenib and dabrafenib inhibitors of B-RAF, which were approved in 2011 and 2013, respectively, for clinical practice in melanoma, whereas MEPK inhibitor trametinib was approved in 2014 for the treatment of advanced BRAF-mutated melanoma, alone and in combination with dabrafenib [101]. MAPK/ERK kinase inhibitor PD-0325901 has also been used in patients with advanced melanoma cases [102].

In this context, the study published by Samaras et al. [98] reveals that p38 MAPK expression had significant association with high Fuhrman’s index in RCC patients, thereby supporting the use of p38 inhibition as a novel therapeutic approach in cancer along with other therapies. However, Carracedo et al. [103] found that prostate cancer cells treated with a MAPK pathway inhibitor escape death through mammalian target of rapamycin complex 1 (mTORC1) pathway in in vitro conditions and xenograft mouse models, as well. Therefore, combining mTOR inhibitor with p38 MAPK inhibitor might have a synergetic effect on cancer cells.

### 8.2. Mammalian Target of Rapamycin Pathway

Another important pathway is phosphoinoside 3 kinase/protein kinase B/mammalian target of rapamycin (PI3K/AKT/mTOR) cascade, which is constitutively activated in various cancers. This pathway is activated by loss of tumor suppressor PTEN function, amplification or mutation of PI3K, amplification or mutation of Akt, activation of growth factor receptors, and exposure to carcinogens [104]. Interruption of this pathway results in antiproliferation, antisurvival, antiangiogenic and proapoptotic effects.

mTOR is a 289 kDa serine/threonine kinase that belongs to the PI3K-related protein kinase (PIKKs) family [70]. It encompasses two functionally distinct protein complexes: mTOR complex 1 and mTOR complex 2 [105,106]. The mTORC1 consists of mTOR, raptor, mLST8, and two negative regulators, PRAS40 and DEPTOR [107]. Raptor regulates mTOR activity and functions by recruiting substrates. Gwinn et al. [108] suggested that mTORC1 action can be regulated by the phosphorylation status of raptor. Another key molecule is mLST8, which positively regulates the mTOR kinase activity while negatively regulates the Tuberous Sclerosis Complex 1/2 (TSC1/2). mTORC1 is the key regulator of ribosomal biogenesis and protein synthesis through ribosomal S6 kinase (S6K), and phosphorylation and inactivation of the repressor of mRNA translation of eukaryotic translation initiation factor 4E-binding protein 1 (4EBP1) [109]. mTORC2 is activated by various growth factors, which phosphorylate protein kinase C alpha (PKC-α), Akt (on Ser473) and paxillin (focal adhesion-associated adaptor protein), and further regulates the activity of the small GTPases Rac and Rho related to cell survival, migration and regulation of the actin cytoskeleton [110].

The mTOR pathway has particular relevance to RCC as it has been shown that expression of HIF is dependent on mTOR activity. Any alteration in the *VHL* gene results in accumulation of HIF-1α and HIF-2α in the majority of cRCC, which is proven to be a critical step in RCC tumorigenesis. A study has shown that phospholipase conjugated mTOR activation enhanced HIF-1α and HIF-2α in RCC [111]. Studies have already shown that temsirolimus treatment impaired expression of HIF-1α under both hypoxic and normoxic conditions in mice bearing RCC xenograft models. This presented another potential mechanism of action for the rapalogs in patients with RCC [112].

Rapamycin, an inhibitor of mTOR and a natural product derived from Streptomyces hygroscopicus, was discovered previously as an antifungal antibiotic from soil on Rapa Nui (formerly called Easter Island). Despite its significant anti-tumoral activity, rapamycin could not be developed for cancer treatment due to its poor aqueous solubility and chemical stability. Subsequently, new rapamycin analogs with improved properties were developed for clinical trials. Currently, Sirolimus, Everolimus (RAD001) and temsirolimus are the three analogs used in practice. Temsirolimus is a soluble ester analog of rapamycin, and has been selected for cancer treatment in preclinical studies based on its anti-tumor, pharmacologic and toxicological characteristics [113]. It was further approved by FDA in 2007 for the first-line treatment of poor-prognosis patients with advanced RCC [114].

By inhibiting mTOR signaling, temsirolimus inhibits the translation of the mRNA that encodes proteins required for G1 progression and S-phase initiation, and that mediate cell growth and angiogenesis [115]. Rapamycin and its analogs partially inhibit mTOR by inactivating mTORC1, but not mTORC2 [106]. Though rapalog therapies have shown clinical efficacy in a subset of cancers [116], this mode of drug action does not fully exploit the anti-tumor potential of mTOR pathway targeting. Additionally, mTORC1 can also negatively regulate PI3K or extracellular signal regulated kinase (ERK)/mitogen-activated protein kinase (MAPK), implicating potential feedback activation of PI3K and/or ERK/MAPK signaling by rapamycin in certain cancer cells [103]. Dual inhibitors of mTOR that inhibit both mTORC1 and mTORC2 complex such as KU0063794, sapanisertib (codenamed INK128), AZD8055, and AZD2014 have now entered clinical trials [117]. They inhibit the phosphorylation of S6K1, 4E-BP1, downstream substrates of mTORC1 and Akt phosphorylation [118].

In recent years, a combination of target therapies has been used for better outcome; this involves either vertical or horizontal inhibition of signaling pathways. The concept of vertical inhibition involves one or more agents that inhibit a specific pathway, useful in addressing the issue of negative feedback loops. Horizontal inhibition involves the use of targeted agents that inhibit two or more signaling pathways. In this context, Lasithiotakis et al. [119] have used a combination of MAPK and mTOR inhibitors, which induce cell death, and abrogate invasive growth of melanoma cells. Another recent study has reported that a combination of temsirolimus and tivozanib has a better effect on RCC patients [120]. A recent study also demonstrated that a combination therapy with CI-1040 and RAD001 could inhibit the growth of Gallbladder cancer (GBC) both in vitro and in vivo. The inhibition of MAPK and mTOR signals affected the expression of several cell cycles and apoptosis-related proteins in GBC cells and induced G1 cell cycle arrest and caspase-dependent apoptotic cell death [121]. Another study by Nakamura et al. [122] has reported that the combination of rapamycin and MAPK inhibitors enhances the growth inhibitory effect of malignant fibrous histiocytoma. Another study showed that mTOR inhibition reduces the cell growth but does not induce apoptosis in target populations but when mTOR inhibition is used in combination with MEK inhibition it reduces growth and induces apoptosis [123]. The MEK inhibitor Selumetinib alone showed no improvement in progression-free survival in patients with soft-tissue sarcomas, whereas its combination with temsirolimus showed clinically meaningful activity in leiomyosarcoma [124]. Moreover, a combination of MEK inhibitor, MEK162 and PI3K inhibitor and BYL719 is undergoing phase II clinical trials for patients with colorectal cancer and esophageal cancer [125], whereas a combination of Everolimus, vinorelbine and trastuzuma is being used in phase III randomized trials of mTOR inhibitors in metastatic breast cancer [126].

To date, there are very few therapies available which are infrequently used to treat RCC, however, no biomarker is known to provide diagnosis at early stages due to low sensitivity and specificity. Therefore, the RCC is considered as one of the most lethal cancers. The information in this context is still lacking with regard to the precise mechanism through which RCC initiates and progresses. To date, a number of therapeutic regimens have been attempted but there is a paucity of information with regard to combination target therapies, especially of inhibitors. Therefore, a study should be designed to evaluate the combined effects of MAPK and mTOR inhibitors in the case of RCC (Figure 1) [127].

## 9. Conclusions

The importance of mTOR as a therapeutic target in RCC has been confirmed by various studies where different mTOR inhibitors, temsirolimus (CCI-779) and everolimus (RAD001) have shown anticancer activity in RCC [128,129]. In patients with advanced RCC, CCI-779 showed antitumor activity and survival rate in phase III clinical trials [127]. The FDA also approved the use of mTOR inhibitor CCI-779 in 2007 for treatment of advanced kidney cancer. However, only a subset of patients responds to this therapy. So, identification of tumor types that respond to mTOR inhibitors remains a major issue for the application of mTOR inhibitors in therapy. In some studies, which were carried out in patients with RCC, resistance to the rapamycin derivative CCI-779 was found to be associated with low levels of phospho-protein kinase B (p-Akt) and p70 ribosomal S6 kinase (p-S6K1) [130]. Therefore, researchers have suggested that further studies need to be conducted to find out whether patients with low or negative p-Akt levels should be excluded from treatment with mTOR inhibitors. mTOR inhibition with CCI-779 has shown dysregulation of glucose and lipid metabolism leading to side effects like hyperglycemia, hypophosphatemia, anemia, and hypertriglyceridemia in patients [131]. In order to reduce the side effects and improve the therapeutic efficacy of mTOR inhibitors, the mTOR inhibitors can be combined with other potential inhibitors of the signaling cascade. The present study therefore suggests that the combined effects of mTOR and MAPK inhibition can be used to induce apoptosis and reduce cell proliferation in vitro as well as in vivo, which may help to advance in the treatment of patients with RCC.

## Figures and Tables

**Figure 1 medsci-04-00016-f001:**
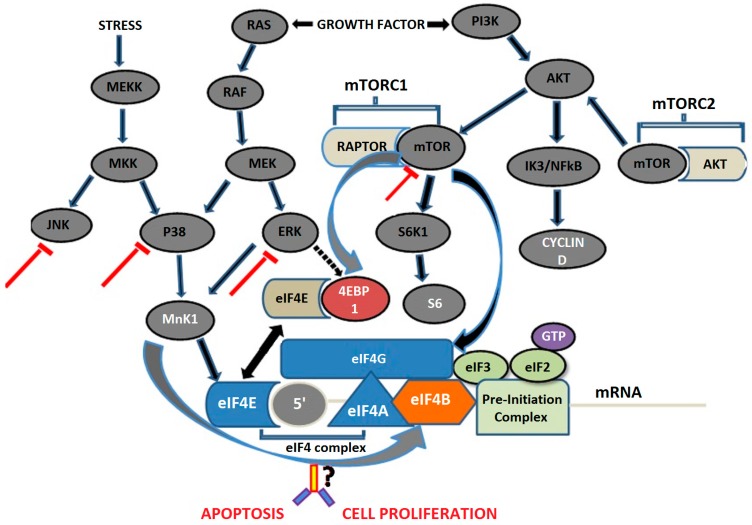
Proposed schematic diagram showing combined inhibition of MAPK and mTOR pathways. [MEKK (MEK kinase); MKK (MAP kinase kinase); JNK (c-Jun N-terminal kinase); MnK1 (MAPK interacting kinase 1); eIF (eukaryotic translation initiation factor); GTP (guanosine triphosphate); S6K1 (S6 kinase 1)].

**Table 1 medsci-04-00016-t001:** Possible risk factors and their role in renal cell carcinoma (RCC).

S. No.	Risk Factor	Finding	Reference
1	Cigarette Smoking	Cigarette smoking has been considered as the most consistent causal risk factor for RCC which accounts for up to 30% of RCC in men and 20% in women.	[21,22]
		A meta-analysis from 19 case-control studies on 1,457,754 participants with 1,326 RCC, revealed a relative risk of RCC as 1.54 in male and 1.22 in female smokers	[23]
		There is significant relationship between smoking and incidence rate of RCC	[24]
		Higher risk of RCC is associated with heavier smoking	[25]
		Quitting smoking reduces the risk of RCC	[26]
2	Alcohol Consumption	There is a positive correlation between kidney cancer and consumption of alcohol	[21]
3	Obesity	Excessive weight is a risk factor for RCC in several patients	[27]
		The proportion of RCC attributable to overweight is estimated to be >40% in USA whereas >30% in Europe	[28,29]
		Obesity with hypertension may lead to increase in lipid peroxidation, which forms DNA adducts and ultimately leads to RCC	[30]
4	Hypertension	There is a relationship between RCC and blood pressure, and people with high blood pressure have higher tendency for RCC	[31,32]
5	Diet	Diet plays a key role in etiology of RCC but investigations have shown no protective effect of vegetables and/or fruits consumption on RCC	[32,33]
		Epidemiologic study suggested high protein consumption as a risk factor for RCC as well as an inducer of renal tubular hypertrophy	[34]
6	Acquired Cystic/Chronic Dialysis	Acquired renal cystic disease develops in patients mostly with end-stage renal disease, and long-term haemodialysis increases the chance of RCC	[35]
		The incidence of RCC in acquired renal cystic disease is found to be 3–6 times higher than that of other cases	[36]
		Long term use of dialysis is reported to be associated with higher incidence of RCC	[37]
7	Inherited Susceptibility	Genetic diseases such as von Hippel-Lindau syndrome (VHL), hereditary papillary renal carcinoma, tuberous sclerosis, Birt-Hogg-Dubé syndrome (BHD) and hereditary leiomyoma are also associated with RCC	[38,39,40,41]
8	Additional Risk Factor	Urinary tract infection	[26]
		Low physical activity	[42]
		Dialysis treatment also increases the risk for developing RCC	[24]
		Radiations appear to increase the chances of occurring RCC	[21]
		Kidney stones increase risk of RCC	[43]

**Table 2 medsci-04-00016-t002:** Primary Tumor (T) Stages in Renal Cell Carcinoma.

TX	T Cannot Assess.
T0	No evidence.
T1	T ≤ 7 cm in greatest dimension, limited to kidney.
T1a	T ≤ 4 cm in greatest dimension, limited to kidney.
T1b	T > 4 cm but not >7 cm in greatest dimension, limited to kidney.
T2	T > 7 cm in greatest dimension, limited to kidney.
T2a	T > 7 cm but ≤10 cm in greatest dimension, limited to kidney.
T2b	T > 10 cm, limited to kidney.
T3	T extends into major veins or perinephric tissues but not into the ipsilateral adrenal gland and not beyond Gerota fascia.
T3a	T grossly extends into the renal vein or its segmental branches, or T invades perirenal and/or renal sinus fat but not beyond Gerota fascia.
T3b	T grossly extends into the vena cava below the diaphragm.
T3c	T grossly extends into vena cava above diaphragm or invades wall of vena cava.
T4	T invades beyond Gerota fascia (including contiguous extension into ipsilateral adrenal gland).

**Table 3 medsci-04-00016-t003:** Regional Lymph Nodes (N) in Renal Cell Carcinoma.

NX	N Cannot Assess.
N0	No N metastasis.
N1	Metastases in N.

**Table 4 medsci-04-00016-t004:** Distant Metastasis (M).

M0	No Metastasis (M).
M1	M appears

**Table 5 medsci-04-00016-t005:** Anatomic Stage/ Prognostic Groups for Renal Cell Carcinoma.

Stage	T	N	M
I	T1	N0	M0
II	T2	N0	M0
III	T1 or T2	N1	M0
	T3	N0 or N1	M0
IV	T4	Any N	M0
	Any T	Any N	M1

Source: American Joint Committee on Cancer (AJCC) Cancer Staging Manual, 2010 [45].

**Table 6 medsci-04-00016-t006:** Distant Metastasis (M) Stages in Renal Cell Carcinoma.

Grade	Nucleus	Nuclear Size (μm)	Nucleoli
1	Round, uniform	10	Absent/inconspicuous
2	Slightly irregular	15	Evident
3	Very irregular	20	Large and prominent
4	Bizarre & multilobated	>20	Prominent, chromatin clumped

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
