# Peer review of "Combination of mTOR and MAPK Inhibitors—A Potential Way to Treat Renal Cell Carcinoma"

_medsci, 2016, doi:10.3390/medsci4040016_

Reviewer 1 Report

This is an interesting review about renal cell carcinoma. However, I found the title imprecisely addressed the content of the paper: mTOR and MAPK are discussed in only 2 pages out of 18 pages.

I strongly suggest to reduce the introductory paragraphs by conveying these informations just through a table: specifically, I suggest to prepare a table of the risk factors together with the Odds ratio (or other quantification of risk) and references, and reduce the entire paragraph 3 by making reference to this table.

I also suggest to discuss the influence of mTOR and MAPK from the clinical point of view: the authors should summarize in a table the clinical trials (and their outcomes) that support an intervention on these two supposed targets. otherwise the evidences they provide remain quite speculative and unsupported.

Author Response

Thank you for your careful review on our paper. We found your comments very helpful in improving our work. The response against your comments is given as following;

Comment 1. I found the title imprecisely addressed the content of the paper: mTOR and MAPK are discussed in only 2 pages out of 18 pages.

Response: We agree with your statement. As till the date, the literature about combination of MAPKs and mTOR is very limited. We included few recent reports on combined treatment with mTOR and MAPK inhibitors in the revised manuscript and the same is highlighted with yellow font.  

Comment 2. Reduce the introductory paragraphs by conveying these informations just through a table: specifically, I suggest to prepare a table of the risk factors together with the Odds ratio (or other quantification of risk) and references, and reduce the entire paragraph 3 by making reference to this table. 

Response: Table 1 is prepared with risk factors and their possible role in RCC.

Comment 3. The influence of mTOR and MAPK from the clinical point of view: the authors should summarize in a table the clinical trials (and their outcomes) that support an intervention on these two supposed targets. otherwise the evidences they provide remain quite speculative and unsupported.

Response: Thank you for raising this comment on clinical trials. The inhibitors of mTOR and MAPK undergoing clinical trial have been incorporated in the text, and highlighted with yellow font.

Reviewer 2 Report

This review needs major revisions. What was the purpose of the review? To give an overview of the  pathogenesis, diagnosis and treatment of renal cell cancer? Because 6 pages are about these subjects with no real line in it and in my idea this is not really important when you want to talk about a combination treatment with mTOR and MAPK inhibitors. It is not clear how he review is performed, what were the criteria to search in the literature? Which articles are used and why,  and how were they selected?

 A lot of the statements about renal cell cancer are really not up to date  (on page 9 for example)” immunomodulatory therapy with cytokines is presently the treatment regimen for RCC”(and then nothing about nivolumab but only some old comments on INF alfa and IL-2.  And on page 11. “till date there are neither therapies available to treat RCC successfully  norbiomarkers available”  ?? this is absolutely not correct in 2016 with sunitinib pazopanib, axitinib, everolimus, nivolumab.

 The presented data about mTOR and MAPK inhibitors are a compilation of all short reports from other articles but there is no binding or line in the story. So I’m not convinced why I should use a combination of these medicines for a future study in mRCC.

The English needs also major revision.

Author Response

Thank you for your careful review on our paper. We have revised the work as per your comments/ suggestion and the changes made in the MS are given in yellow font.

Comment 1. What was the purpose of the review? To give an overview of the pathogenesis, diagnosis and treatment of renal cell cancer? Because 6 pages are about these subjects with no real line in it and in my idea this is not really important when you want to talk about a combination treatment with mTOR and MAPK inhibitors. It is not clear how he review is performed, what were the criteria to search in the literature? Which articles are used and why, and how were they selected?

Response: Following phrase is added to the introduction part;

The purpose of the present review is to explore the current status, particularly pathogenesis, diagnosis and treatment of RCC. The literature was imported from PubMed and Scopus data base by searching terms “mTOR inhibitor and RCC”, “MAPK inhibitor and RCC”, “combination of mTOR and MAPKs inhibitors”, “renal cell carcinoma and clinical trial”, etc. Around 500 relevant papers were review in which only most relevant reports were considered. This review emphasizes information from prevalence to challenges in the treatment of RCC, and its solution with the help of combination target therapy.

Comment 2. A lot of the statements about renal cell cancer are really not up to date  (on page 9 for example)” immunomodulatory therapy with cytokines is presently the treatment regimen for RCC”(and then nothing about nivolumab but only some old comments on INF alfa and IL-2.  And on page 11. “till date there are neither therapies available to treat RCC successfully  norbiomarkers available”  ?? this is absolutely not correct in 2016 with sunitinibpazopanib, axitinib, everolimus, nivolumab.

Response: Thank you for raising this important comment. We have corrected the issues accordingly and highlighted with yellow font.

Comment 3. The presented data about mTOR and MAPK inhibitors are a compilation of all short reports from other articles but there is no binding or line in the story. So I’m not convinced why I should use a combination of these medicines for a future study in mRCC.

Response: As there are many diagnostic tools available for detection of RCC but none of them detect disease at early stage. However, at later stage of the disease, there is no therapy which treats successfully. Some cases of RCC respond to mTOR target therapy and some of them show remission due to alternate pathway to escape from death. So, the present review proposed a combination of mTOR and MAPKs therapy for better outcome.Response: As there are many diagnostic tools available for detection of RCC but none of them detect disease at early stage. However, at later stage of the disease, there is no therapy which treats successfully. Some cases of RCC respond to mTOR target therapy and some of them show remission due to alternate pathway to escape from death. So, the present review proposed a combination of mTOR and MAPKs therapy for better outcome.

Reviewer 3 Report

The authors have spent most part of the article on the background, with the therapy part coming in 3 pages (page 10-12). Even at that time, It is under the subtitle "8. Molecular pathways in RCC", with no mentioning of mTOR and MAPK inhibitors.  It is too much of "hide and seek".

The article is more of "Perspective" than a "Review", because no one has done the combination of mTOR and MAKP inhibitors for RCC.

Despite that, this is still a useful article to some readers of this journal.  The proposal of the combination by the authors is potentially a good way to treat RCC.

Some minor revisions are recommended. In addition to the issues raised above, there are some typos in the manuscript.  For example, (1). Line 432: "rapamycin could not developed cancer treatment" where is the verb?  (2). Line 466: norbiomarkers needs a space between the two words.

Author Response

Thank you very much for your constructive comments to improve our manuscript. Your comments and our response are given below one by one.

Comment 1. Molecular pathways in RCC", with no mentioning of mTOR and MAPK inhibitors.

Response: We carefully revised the manuscript and more information related to mTOR and MAPK inhibitors as well as undergoing clinical trial have been incorporated. Changes are highlighted with yellow font.

Comment 2. The article is more of "Perspective" than a "Review", because no one has done the combination of mTOR and MAKP inhibitors for RCC.

Response: There are very few reports available in evidence of combination therapy for RCC. We included these reports in the manuscript including a thesis of A. Chauhan at ref. no. 127 as given below.

Chauhan A. Evolution of Speckle-type POZ protein (SPOP), a biomarker and its inhibition in the combination therapy in renal cell carcinoma in vitro. Thesis: Postgraduate Institute of Medical Education & Research, Chandigarh, India, July, 2014.

Comment 3. Line 432: "rapamycin could not developed cancer treatment" where is the verb?  (2). Line 466: norbiomarkers needs a space between the two words.

Response: Lines are corrected accordingly.

Round  2

Reviewer 1 Report

the authors have asnwered all my suggestions

Reviewer 2 Report

I think that the title and abstract do not really cover the review article and the statement that mTOR and MAPK inhibitors are a potential way of treatment for RCC is so much based on speculation that there is not enough ground to make such a big article on it.

On my comment 2. that a lot of the statements about renal cell cancer are really not up to date. Those sentences are still in the article. although some sentences are made about new treatments.